# Using citation network analysis to enhance scholarship in psychological science: A case study of the human aggression literature

Alessia Iancarelli[1]*, Thomas F. Denson[2], Chun-An Chou[3], Ajay B. Satpute[1]

**1** Department of Psychology, Northeastern University, Boston, MA, United States of America, **2** School of Psychology , University of New South Wales, Sydney, NSW, AUS, **3** Mechanical Industrial Engineering, Northeastern University, Boston, MA, United States of America

* iancarelli.a@northeastern.edu

**Data Availability Statement:** The data underlying the results presented in the paper are available from the open access Semantic scholar database. The code to extract the data is available at: https://

## Abstract

Researchers cannot keep up with the volume of articles being published each year. In order to develop adequate expertise in a given field of study, students and early career scientists must be strategic in what they decide to read. Here we propose using citation network analysis to characterize the literature topology of a given area. We used the human aggression literature as our example. Our citation network analysis identified 15 research communities on aggression. The five largest communities were: "media and video games", "stress, traits and aggression", "rumination and displaced aggression", "role of testosterone", and "social aggression". We examined the growth of these research communities over time, and we used graph theoretic approaches to identify the most influential papers within each community and the "bridging" articles that linked distinct communities to one another. Finally, we also examined whether our citation network analysis would help mitigate gender bias relative to focusing on total citation counts. The percentage of articles with women first authors doubled when identifying influential articles by community structure versus citation count. Our approach of characterizing literature topologies using citation network analysis may provide a valuable resource for psychological scientists by outlining research communities and their growth over time, identifying influential papers within each community (including bridging papers), and providing opportunities to increase gender equity in the field.

## 1 Introduction

The volume of published literature has been rapidly growing over the past several decades [1, 2]. In tandem, the competition for academic positions is also increasing, and so students are often advised to publish more articles to be competitive for these positions [3, 4]. By one count, assistant professors who were hired in the 2012–2016 cohort had 57% more published articles in their records than the 2006–2011 cohort [5]. This increasing growth of science leads to a new challenge particularly for students and early career researchers. Simply put, how are scientists to develop their scholarship in a field given the volume of research that has been

github.com/ABS-Lab/A-Network-Approach-to-the-Human-Aggression-Literature.

**Funding:** The author(s) received no specific funding for this work.

**Competing interests:** The authors have declared that no competing interests exist.

published to date and that continues to be published each year? It is important for researchers to develop a clear strategy in selecting their readings. One possible strategy is to select papers on the basis of total citation count [6]. This approach may be useful for a field if it is relatively homogenous in scope. However, if a field is composed of multiple communities, focussing on the total citation count alone is likely to introduce biases in scholarship. For instance, it will likely lead to over-representation of the most popular community of the field and under-representation of other important research communities. This approach may also contribute to field-wise biases in who gets cited by whom [7], thereby leading to systematic disparities in citations, for example, by gender [8]. Here, we approach the question of scholarship development using citation network analysis [6, 9–14]. We assayed the human aggression literature as our case study. According to Semantic Scholar, over 46,000 peer-reviewed articles on aggression have been published since 1932. If we focus solely on human aggression, about 7,400 journal articles have been published in the last two years alone. Thus, like many growing fields, there is a considerable barrier to entry for establishing breadth and depth of scholarship. By applying citation network analysis to this literature, we illustrate its utility for mapping the research topology on human aggression. We identify subfields in the aggression literature, influential papers (based on graph theoretic measures) for each subfield, and bridging papers that connect and temporally antecede subfields. We further examine whether citation network analysis reduces gender disparity by highlighting as influential more papers from women authors.

## 1.1 Citation network analysis

Citation network analysis uses graph theory to model literature based on which articles cite one another. It treats papers as "nodes" in the graph, and citations between papers as "edges" in the graph. Previous work has used this approach to explore diverse scientific fields including radio frequency identification [15], sport burnout [16], human resource development [17], technology management [18], and more (e.g., Benckendorff et al.(2013) [19]; Dawson et al. (2014) [20]; Ebesu, & Fang (2017) [21]). By and large, prior work has focussed on various measures of network centrality [15, 16, 22]—which captures how centrally positioned a paper is in the network (e.g., based on whether many papers cite that paper, whether the closest path between two papers involves that paper, etc). Centrality measures are well-suited for identifying particular papers that may be influential for the network structure. However, they do not, on their own, provide information about whether a field is relatively homogenous or whether it is composed of multiple distinct communities. Here, clustering methods can be used to group sets of papers into communities. For example, Jeung (2009) [17] found that the human resources development literature contains four main research communities based on how similar the papers were to one another with respect to their citation profiles. In this article, we take advantage of both approaches. We combine graph theoretic measures for specific papers (e.g., centrality) and also measure that group papers together (e.g. using clustering methods) in our citation network analysis. Given a citation matrix, our analysis will identify a set of communities, how large the communities are, and how they are connected to one another (i.e., "bridge papers"). It will also indicate which specific papers are more influential in each community (at least from a graph theoretic perspective), and thus, provide a potential reading list that can be used to develop expertise across multiple communities of a field. We also tabulate our results by the dates of publications and citations to descriptively explore the growth of communities over time.

## 1.2 Application to research on human aggression

We applied our citation network analysis to the research literature on human aggression. Aggression is of tremendous societal importance. Indeed, there has been an escalation of aggression associated with society traumatic events including, but not limited to, police violence and the Black Lives Matter movement, COVID-19 and violence against East Asians, and political upheavals and violence across the political divide. Domestic violence [23–27] including violence against children [26, 27] has also increased during COVID-19, as have factors associated with increased aggression (e.g., alcoholism [28]). A citation network analysis applied to the aggression literature may help speed scholarship training and future research in the field of aggression by helping scholars identify certain research communities and the influential articles of each community.

The interpretation of any citation network analysis will depend on the research articles that are included in the network. We took a seed-based approach in which the citation matrix is generated from articles that reference a landmark seed article and from articles that reference those articles. Selection of an appropriate seed—one that is highly connected with the broader aggression field—will likely recapitulate the most robust properties of the aggression literature at large while also ensuring that all articles of the network are connected to one another. Seed papers can be selected via quantitative criteria (e.g., citations) or by using qualitative criteria (e.g., the publication outlet, particular research teams) or a combination of both criteria [29]. In the present research, we selected our seed paper based on both qualitative and quantitative information. This hybrid approach confers important advantages for conducting citation network analysis on the aggression field as we outline below.

For our seed, we selected a landmark Annual Review of Psychology article titled 'Human Aggression' by Anderson and Bushman (2002) [30]. We selected it on the basis of several criteria. First, it is the most cited peer-reviewed review article on human aggression. Second, it is also a relatively modern article, which will usefully introduce a bias in our citation matrix to identify research communities that are of relevance to contemporary research on aggression. Third, articles published in the journal Annual Review of Psychology are written primarily for a research audience and with the intent of providing a comprehensive review of the field. Naturally, they serve as landmark articles which are cited by researchers from many different communities in the field. While there are other highly influential contributions on aggression (notably, books by Storr (1968), Geen (1998), and Baron (2004) [31–33]), these were published decades earlier, were not peer-reviewed, focussed on a particular perspective rather than providing a more comprehensive overview, and were intended for a more general audience rather than focussing more specifically on the research community. Thus, it stands to reason that the Anderson and Bushman (2002) [30] article would serve as an optimal seed for our purposes. From our seed article, we developed our citation matrix by including two generations of articles—that is, 1st generation articles that cited the seed article, and for greater breadth, 2nd generation articles that cited the 1st generation articles (many of which may have also cited the seed article), which we then submitted to our citation network analysis.

Apart from the seed selection step, our approach is inherently data-driven and exploratory. Even so, we anticipated that our approach would identify several research communities and would identify many influential articles that otherwise would have been overlooked if focussing only on total citation count. Like most research areas, the field of aggression has already been divided into several topics as evidenced by distinct symposia at relevant conferences. Our seed based approach also provides a useful opportunity to validate our methods. The review article identified several major topics in the field such as alcohol and aggression, aggression and video games, etc. Here, we see whether our data-driven method will recapitulate

aggression research communities as characterized by experts in the field. Critically, the seed article could only concern articles from the past, whereas our approach focuses on articles published after the seed article. Thus, our approach also identifies new and emerging communities in the field, and further, how interconnected these communities are with one another.

## 2 Methodology

In network analysis, a network is composed of nodes and edges that connect nodes. In our analysis, each node refers to a single paper, and the edges between nodes refer to whether that paper is cited by another paper. For instance, if paper A is connected through an edge to paper B, it means that paper A is cited by paper B. Thus, our human aggression network is a directed (unweighted) graph, which means that the edges have a specific direction. Fig 1 shows a visual example of a citation network.

### 2.1 Paper gathering

From our source paper, Human Aggression [30], we obtained first generation and second generation articles, or papers that directly cite the source and papers that directly cite the first generation articles, respectively. This method ensures that the papers collected are already linked through citations of each other in some fashion, thus ensuring there will be sufficient edges for our analysis. Papers and their relevant attributes were gathered using the application programming interface (API) offered by the free platform Semantic Scholar. The project was developed principally using the Python programming language [34]. In total, we obtained 7,862 papers. Of these, many papers had very few citations, and therefore would not significantly contribute

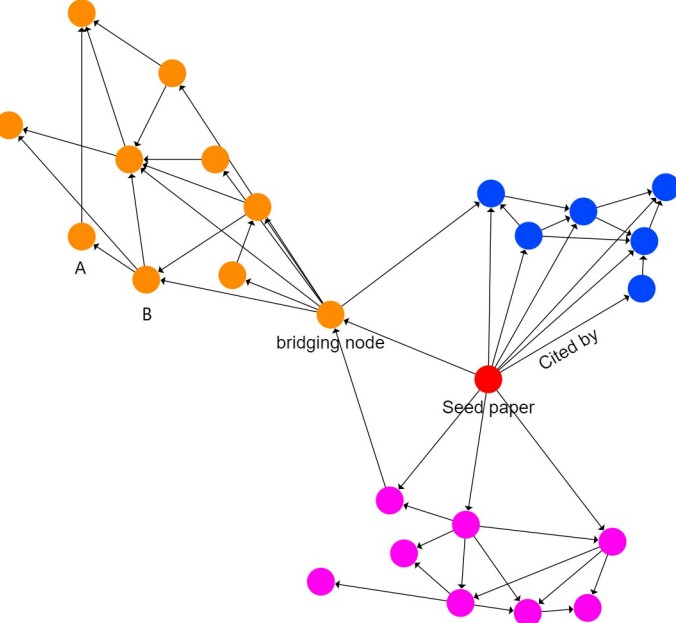

**Fig 1. Example of a directed unweighted citation network.** The network consists of 25 nodes and 50 edges organized in three communities (colored in blue, orange, and fuchsia). The seed paper (source article) is colored in red. Every node represents a paper whereas every edge between nodes refers to whether that paper is cited by another paper (as indicated by the label "cited by"). An example of bridging node is signaled by the label "bridging node". Bridging papers serve as links between two or more communities in the network.

to identifying network structures. Hence, we filtered out papers with two or fewer total citations resulting in 1,345 nodes (papers) and 5,148 edges.

## 2.2 Citation network analysis

We used citation network analysis to search for communities in the human aggression literature. We utilized a seed-based approach, where the network is generated by including all the papers (nodes) that cite the seed paper, and, in our case, all the papers which cite those (two levels of distance from the seed). There were four main aspects to our citation network analysis. First, we implemented the widely used "Louvain community detection algorithm" (MATLAB Brain Connectivity Toolbox [35]). There exists a considerable amount of community detection algorithms, yet, the Louvain algorithm was preferable in our case because it includes a community aggregation step to improve processing on large networks [36]. Conceptually, "communities" are formed from groups of papers that tend to be cited by the same papers. This algorithm maximizes a modularity score for each community; specifically, it compares how much more densely connected the nodes within a community are with how connected the nodes would be in a random network [36].

Second, we identified papers that seemed most influential in terms of certain network metrics (the metrics are shown in Table 1 and are described further in the Results section). Although each metric attempts to characterize different properties of nodes in a network, we conducted a principal components analysis of these metrics and found that all but one of the network metrics loaded onto a single component which accounted for nearly 59% of the variance in the measures (see S1 File). To be sure, high correspondence between different graph theoretic measures is not a given; rather, it is a function of the input matrix. Since each paper was at most two degrees

**Table 1. Metrics.** In addition to the metrics reported in this Table, the Density Maximum Neighborhood Component (DMNC) metric, which detects densely connected neighborhoods, was calculated. However, DMNC was not taken into account for further analysis (see Section D in S1 File).

| Metric | Substantial Description | Mathematical Description | Mathematical equation |
|---|---|---|---|
| Betweenness centrality (Btw) | Bet counts the number of times a node lies on the shortest path between other nodes, and it shows which nodes are 'bridges' between nodes in a network | Btw of a node is defined by the number of shortest paths between any couple of nodes that pass through that particular node. | $BtW(v) = \sum\limits_{s \neq t \neq v \in C(v)} \dfrac{\sigma st(v)}{\sigma st}$ (1) |
| Degree centrality (DC) | DC reveals which nodes have a high number of connections with other nodes. | DC of each node $v$ in a network is calculated by counting the number of edges that node $v$ shared with $m$ other nodes within the network. | $DC(v) = |N(v)|$ (2) |
| Closeness centrality (CC) | Given a node $v$, CC measures how short the shortest paths are from a node $v$ to all nodes. | Given the length of the shortest paths between a given node and all other nodes in the graph, CC is equal to the reciprocal of the sum of these paths. | $CC(v) = \sum\limits_{\omega \in V} \dfrac{1}{dist(v, \omega)}$ (3) |
| Bottle Neck (BN) | BN nodes play key roles in mediating communication within a given network because they facilitate information flow from densely connected sub-networks (Charitou et al., 2016). | For every node $n$ in a network, it is built a tree ($T_n$) of shortest paths that starts from $n$. A node $w$ is defined as BN in $T_n$ if its weight is no less than $\frac{m}{4}$, where $m$ is the number of nodes in $T_n$. The score of node $w$, BN ($n$), is equal to the number of nodes $n$ such that $w$ is a BN node in $T_n$. | $BN(n) = \sum\limits_{s \in n} P_s(n)$ (4) |
| Edges Percolated Component (EPC) | EPC indicates which nodes are important to keep the structure of the graph in place. If these papers are removed, a great amount of nodes may become isolated. | Given a threshold ($0 \leq$ the threshold $\leq 1$), 1000 reduced networks are created by assigning a random number between 0 and 1 to every edge and remove edges if their associated random numbers are less than the threshold. | $EPC(v) = \dfrac{1}{|V|} \sum\limits_{k=1}^{1000} \sum\limits_{t \in V} \delta_{vt}^k$ (5) |
| Maximum Neighborhood Component (MNC) | MNC reveals which nodes have a highly connected node within their neighborhood. | The neighborhood of a node $v$, creates a sub-network $N(v)$. The score of node $v$, $MNC(v)$, is equal to the size of the maximum connected component of $N(v)$. The neighborhood $N(v)$ is the set of nodes adjacent to $v$ and does not contain node $v$ | $MNC(v) = |V(MC(v))|$ (6) |

away from the seed paper, it will likely increase the correspondence between many of these graph theoretic measures. Thus, we used the component score from the principal component as a composite measure of each node's influence in the network. Papers with high composite scores were those that were also high on all of the measures included in Table 1.

Third, we aimed to identify the papers that serve as links between two or more communities in the network. To do so, we used the metric "Bridging Centrality." Conceptually, if there are two (or more) nodes from different communities that are highly connected to other nodes within their respective communities, a node that connects these two (or more) nodes to one another would have a high Bridging Centrality score. That is, we use the metric "Bridging Centrality" to detect nodes that link together two or more communities. Bridging Centrality of a node is the product of the Bridging Coefficient (BC) and the Betweenness centrality (Btw) (1) (Bridging centrality = BC * Btw). The Bridging Centrality of a node is high if the node has neighbors with high degree. Given a specific node i, we define a neighbor as a set of all the nodes adjacent to i including i. The formula for BC is reported below.

$$BrCoe(i) = 1 / \sum_{k=1}^{n} dis_{ik} \qquad (7)$$

The BC of node i is the reciprocal of sums of the distance (dis) from it to all other nodes excluding its neighbors and indirectly adjacent nodes [37]. A node can be defined as having high Bridging Centrality if it is a connecting node (Btw component) of high degree nodes (BC component). Fourth, we examined the extent to which communities are interconnected. To do that, we calculated the average shortness distance between each community pair. Subsequently, we inputted the distance matrix, containing the pairwise distances between communities, to Multidimensional Scaling (MDS) in order to visualize the level of similarity between communities. Cytoscape [38] was used for network visualization.

## 2.3 Network validation

In order to validate the community structure of the network, the network architecture was recalculated 2,000 times by applying the community Louvain algorithm to a subset of 90% of randomly selected nodes. Normalized Mutual Information (NMI) was calculated to evaluate the consistency of solutions across the 2,000 repetitions [39]. NMI assigns a value of 0 where two sets of repetitions are completely orthogonal and 1 where there is a perfect overlap between the two.

## 2.4 Gender analysis

We examined the influential papers found in the communities as well as in the composite ranking and citation ranking, to check for potential gender biases. Precisely, we measured the gender percentage within the Top 75 papers in the citation ranking, the Top 75 articles in the composite score ranking, and the Top 5 papers per community (15 communities considered), for a total of 75 papers. To achieve this, we estimated the first authors' gender by relying on names, pictures, and pronouns used to refer to them. Although we acknowledge that gender is a spectrum, we were unable to characterize gender more fluidly given the data we had available and thus had to rely on normative assumptions. We hope our approach will be useful for characterizing contributions from scientists across the gender spectrum in future iterations of this work.

## 3 Results

### 3.1 Community detection

Our first aim was to identify how many communities there are in the aggression research network and their relative sizes. By using the community Louvain algorithm, we detected 21

**Table 2. The titles for the communities with at least 10 papers are reported in this table along with the number of papers contained in each community.** See the Section B in S1 File, for additional details.

|      | Community Title | Number of papers |
|------|------------------|------------------|
| 1.   | media & videogame | 344 |
| 2.   | stress, traits, and aggression | 226 |
| 3.   | rumination and displaced aggression | 198 |
| 4.   | role of testosterone | 117 |
| 5.   | social aggression | 73 |
| 6.   | PTSD | 60 |
| 7.   | supervisor's aggression | 56 |
| 8.   | social pain & exclusion | 54 |
| 9.   | oxytocin | 54 |
| 10.  | injustice | 44 |
| 11.  | alcohol and aggression | 34 |
| 12.  | no title | 31 |
| 13.  | anger | 22 |
| 14.  | literature Reviews guidelines | 11 |
| 15.  | aggression and horses | 10 |

communities in our network (see S1 File). We only retained communities with at least 10 papers for a total of 15 communities (Table 2).

To characterize the research content of each community, we identified the most influential papers within each community using our composite ranking score of network measures from Table 1 (see Methods and S1 File for details on composite score calculation). The titles from the top five papers in each community were then used to heuristically infer the main topics for that community. As shown in Fig 2 and Table 3, the largest communities concerned: Media and Videogames; Stress, Traits, and Aggression; Rumination and Displaced Aggression; Role of Testosterone; and Social Aggression. The remaining communities are summarized in the supplementary materials (Section B in S1 File), which shows the detailed list of the top five articles for the other communities. It is useful to compare our findings from the community detection analysis (which focuses on clusters of papers) with citation ranking and graph theoretic measures (which focus on individual papers) irrespective of community. The top 10 papers based on citation ranking and based on our composite measure of influence are presented in the supplementary material (Section C in S1 File) and Table 4, respectively. Strikingly, with the exclusion of the seed paper, no paper from the second biggest community appears among the top 10 most cited articles nor in the top 10 most influential papers from our composite ranking. Indeed, 7 of the top 10 papers based on citation ranking were all from a single community. Altogether, these findings highlight the usefulness of the clustering approach in uncovering the community-based topology of a research field (Table 2) and our composite ranking score in identifying key articles for each community (Table 3).

## 3.2 Network validation

Our aggression research network had an average (and median) NMI of .64, SD = .0032. This means that there is a good consistency among repetitions in terms of the number of communities and community structure. However, this is not a perfect overlap which implies that some papers will be placed in different communities across repetitions, and some papers will be

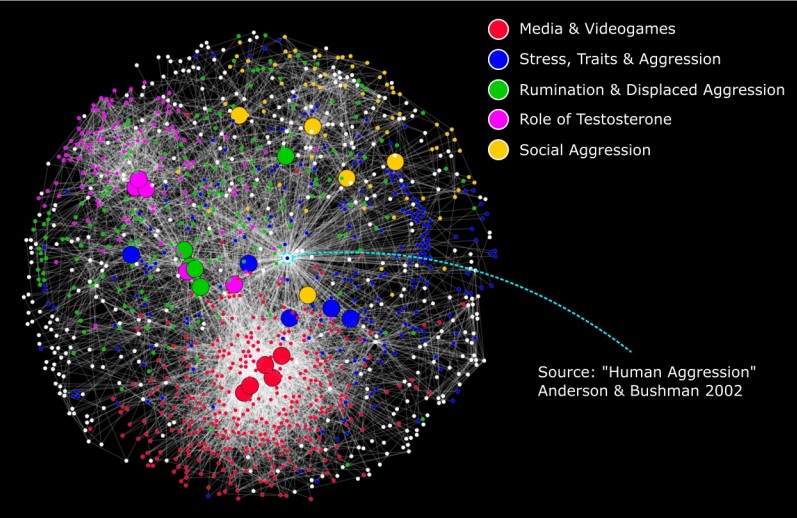

**Fig 2. Top 5 communities in the human aggression network.** (i) Media and Videogames (red), (ii) Stress, Traits, and aggression (blue), (iii) Rumination and Displaced aggression (green), (iv) Role of Testosterone (pink), (v) Social aggression (yellow). The largest nodes represent the five most influential papers within each community.

**Table 3. Top 5 articles for each of the Top 5 largest communities in the human aggression network.**

| | Media & videogame | Stress, traits & aggression | Rumination & displaced aggression | Role of testosterone | Social aggression |
|---|---|---|---|---|---|
| 1 | The Influence of Media Violence on Youth [40] | Human aggression: A social-cognitive view [41] | Chewing on it can chew you up: effects of rumination on triggered displaced aggression [42] | Testosterone responses to competition predict future aggressive behaviour at a cost to reward in men [43] | Atypical empathic responses in adolescents with aggressive conduct disorder: a functional MRI investigation. Biological psychology [44] |
| 2 | The effects of prosocial video games on prosocial behaviors: international evidence from correlational, longitudinal, and experimental studies [45] | Violent video games stress people out and make them more aggressive [46] | Self-regulatory failure and intimate partner violence perpetration [47] | Angry affect and violence in the context of a psychotic illness: A systematic review and meta-analysis of the literature [48] | Psychopathic predators? Getting specific about the relation between psychopathy and violence [49] |
| 3 | Correlates and consequences of exposure to video game violence: hostile personality, empathy, and aggressive behavior [50] | Fueling the fire: Violent metaphors, trait aggression, and support for political violence [51] | The Cognitive Basis of Trait Anger and Reactive Aggression: An Integrative Analysis [52] | Neural Mechanisms of the Testosterone–Aggression Relation: The Role of Orbitofrontal Cortex [53] | Being hot-tempered: Autonomic, emotional, and behavioral distinctions between childhood reactive and proactive aggression [54] |
| 4 | The effect of video game violence on physiological desensitization to real-life violence [55] | Trait Aggressiveness and Situational Provocation: A Test of the Traits as Situational Sensitivities (TASS) Model [56] | The displaced aggression questionnaire [57] | A longitudinal study of the association between violent video game play and aggression among adolescents [58] | A Review and Reconceptualization of Social Aggression: Adaptive and Maladaptive Correlates [59] |
| 5 | Violent video game effects on aggression, empathy, and prosocial behavior in eastern and western countries: a meta-analytic review [60] | Stress leads to prosocial action in immediate need situations [61] | Comm Research—Views from Europe\| Five Challenges for the Future of Media-Effects Research [62] | State, not trait, neuroendocrine function predicts costly reactive aggression in men after social exclusion and inclusion [63] | Testosterone, cortisol, and serotonin as key regulators of social aggression: A review and theoretical perspective [64] |

**Table 4. The Top 10 most relevant papers (excluding the source paper) in our aggression network.** For the top 11-20 see the Section A in S1 File.

| | Titles | Community |
|---|---|---|
| 1. | The Influence of Media Violence on Youth [40] | media & video games |
| 2. | The effects of prosocial video games on prosocial behaviors: international evidence from correlational, longitudinal, and experimental studies [45] | media & video games |
| 3. | Correlates and consequences of exposure to video game violence: hostile personality, empathy, and aggressive behavior [50] | media & video games |
| 4. | The effect of video game violence on physiological desensitization to real-life violence [55] | media & video games |
| 5. | Short-term and long-term effects of violent media on aggression in children and adults [65] | alcohol and aggression |
| 6. | Chewing on it can chew you up: effects of rumination on triggered displaced aggression [42] | rumination & displaced aggression |
| 7. | Violent video game effects on aggression, empathy, and prosocial behavior in eastern and western countries: a meta-analytic review [60] | media & videogames |
| 8. | The effects of reward and punishment in violent video games on aggressive affect, cognition, and behavior [66] | media & videogames |
| 9. | Testosterone responses to competition predict future aggressive behaviour at a cost to reward in men [43] | Testosterone & aggression |
| 10. | Understanding impulsive aggression: Angry rumination and reduced self-control capacity are mechanisms underlying the provocation-aggression relationship [67] | media & videogames |

placed in communities that might not reflect the topic of their titles. Even so, the reasonably high NMI indicates that the overall structure is preserved across repetitions.

## 3.3 Bridging papers

So far, our analysis identifies certain research communities and the most influential papers for each community. It may also be useful to know which papers in a given community serve as bridges to other communities. Bridge nodes may be important for information exchange across communities throughout a network [37]. Conceptually, bridge nodes are papers that are often on the shortest path between papers (i.e. have high betweenness centrality) and that are cited by other papers from different groups (i.e., communities). Importantly, since our citation network is directed, a bridge node is one that is connected through unidirectional links (i.e., "cited by" relationship). In our directed graph, we found that the top five bridging nodes for each community ended up being the same as the five most influential papers of that community (Table 3). To be sure, this overlap does not necessarily reflect an inherent relationship between the influence metric and the bridging metric—it is likely that for other citation network analyses, bridging papers may not overlap with influential papers within a community. However, for newcomers to the human aggression literature, this overlap can be viewed as helpful insofar as it helps constrain the subset of papers that one might select for developing scholarship.

## 3.4 Community interconnectedness

We used a measure of interconnectedness to determine which communities were closer and further from each other. Pairwise interconnectedness scores by communities were submitted to MDS for visualization. As shown in Fig 3, media and videogames, rumination and displaced aggression, and stress traits and aggression appear to be more interconnected with one another, suggesting that scholars are to some extent aware of the work done across these

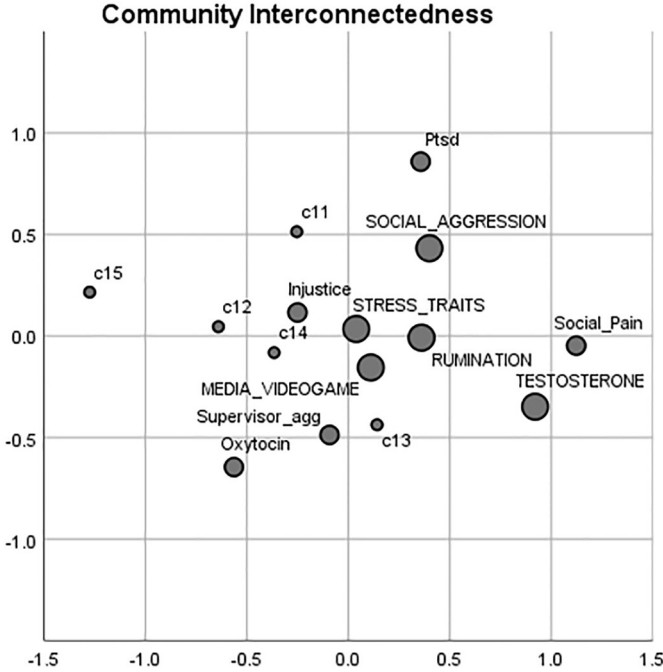

**Fig 3. Community interconnectedness.** Pairwise interconnectedness scores (average shortest path) by communities were submitted to Multidimensional Scaling in order to visualize the level of similarity between communities.

communities. Conversely, social pain and exclusion, testosterone, PTSD as well as oxytocin are relatively more isolated communities. Aggression in horses was the most isolated community.

## 3.5 Community growth

Next, we investigated how communities grew in size and influence over time. That is, we plotted the quantity of publications and the total number of citations across years per community. The boundary of our time window, given our citation matrix, is from 2002–2019 and so interpretation of findings should be understood as contingent on this temporal boundary. Further, interpretation should also be contingent on our seed-based approach of developing the citation matrix. We focus our description primarily on the major trends that are observable from the results. Fig 4 displays the findings. Naturally, the number of citations trailed the number of publications for each community. The three largest communities—"media and video games", "stress traits and aggression" and "rumination and displaced aggression"—have steadily grown over time with peaks around 2015–2016 in volume and around 2017 in total citations. The two smaller of the top five communities—"testosterone" and "social aggression"—appeared to start their growth later in time but peaked at around the same time.

## 3.6 Underrepresentation of women in citation and composite ranking

Methods for developing scholarship in a discipline should also be evaluated with respect to exacerbating gender bias in citations and literature reviews. Indeed, prior work has shown that papers with women authors tend to be cited less frequently than papers with male authors, at least in certain fields [8]. Here, we compared three methods with regard to gender bias. First,

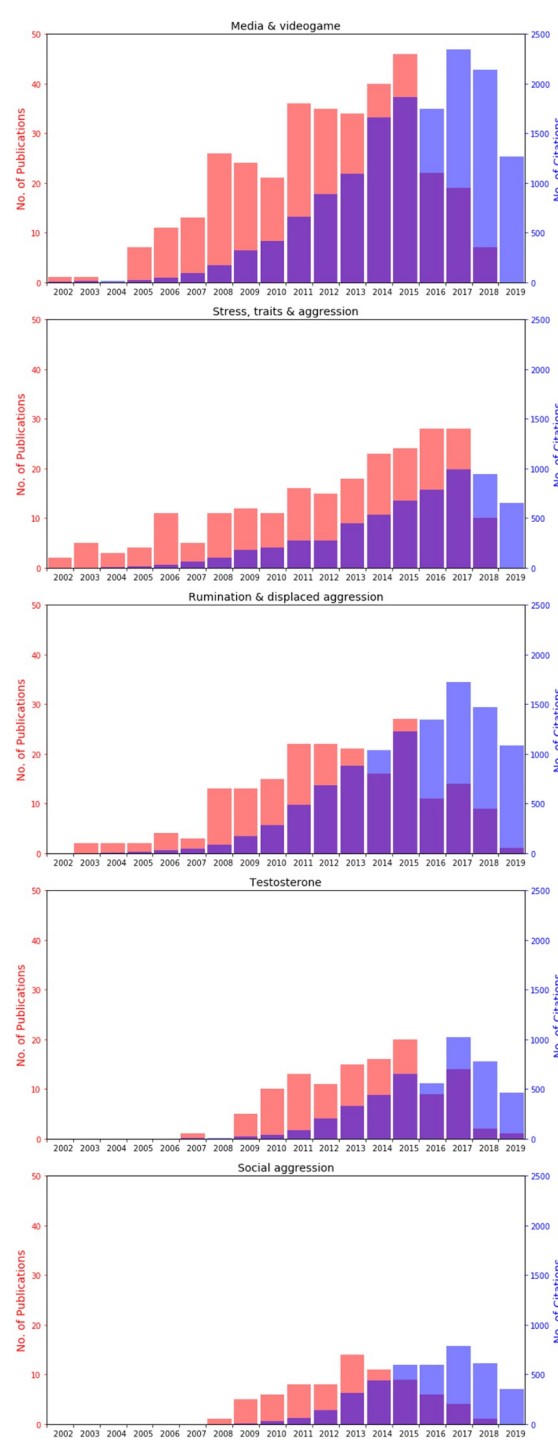

**Fig 4. Community growth.** The bar charts represent the frequency of publications (in red) and number of citations (in blue) per year by community (for the 5 largest communities). On the x-axis are represented the years between 2002 (year of origin of the network: the seed paper was published in 2002) and 2019 (end of data gathering). It is possible to notice that in every community there is a steep decrease of publications and citations in the year 2019. We suspect that this trend might be due to a delay in the uploading of papers in the semantic scholar database and it should not have to reflect a real significant decrease of the interest in the field.

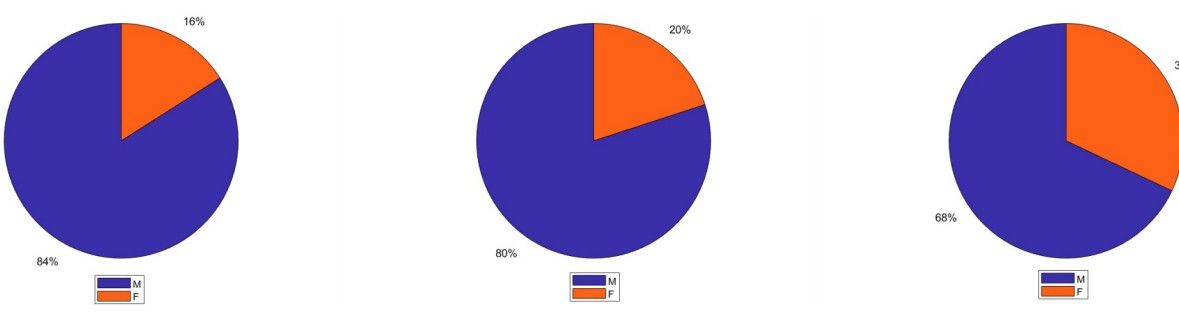

**Fig 5. Gender percentage.** Pie charts of gender percentage (Female and Male) for the correlation ranking (top 75), composite ranking (top 75), and communities (top 5 papers for each community, n 75).

and as a benchmark, we selected the top 75 articles on the basis of citation ranking. Second, we selected the top 75 articles on the basis of our composite influence score (irrespective of the community). Third, we selected 75 articles from our community detection analysis by selecting the top 5 articles from 15 communities. In each case, we counted the proportion of papers with a woman first author. As shown in Fig 5, 16%, 20%, and 32% of the first authors were women for each method, respectively. These findings suggest, first, that there is variation in gender representation based on the approach for developing scholarship, and second, that the community detection approach appears to substantially reduce gender bias in selecting influential articles. Identifying research communities also affords the opportunity to assess which particular communities have an underrepresentation of women first authors. Women were most underrepresented in "media and videogames" and "alcohol and aggression". Both these communities included 0% women (among the top 5 articles for each community). Women were highly represented in research communities on "social aggression", "horses", and "supervisor's aggression", which had 80%, 100%, and 60% women as first authors among the top 5 articles for each community.

We also explored community size. There appeared to be a tendency for larger communities to involve fewer influential publications with women as first authors. For the five largest communities, 28% of the included articles had women as first authors. The middle five communities were similar; 28% of the included articles had women as first authors. Yet, the third group showed the greatest percentage of women as first authors at 40%. However, we note that our sample size (i.e. the number of communities) is too small for a formal analysis.

## 4 Discussion

The volume of publications for any given field in psychology is too vast to be read by a single scientist. In order to acquire a reasonably broad understanding of the field and its many areas of work, researchers must be strategic in selecting their readings. Here, we used citation network analysis to address this issue. We produced the first citation-based literature map of research on human aggression, our topic of interest. Our analysis identified 15 research communities on aggression, how interconnected these communities are with one another, the top five influential papers within each community using a composite of graph-theoretic measures, and papers that bridge communities—which happened to be the same as the most influential papers for each community. In doing so, we provide researchers with a topological map of the aggression literature. We also examined how our citation network analysis would perform in terms of gender representation for publications labeled as influential. We found that the community detection approach—which identifies distinct research communities and then

identifies influential articles within each community using our composite score—doubles the proportion of articles with women as first authors that are deemed as influential. While we applied citation network analysis to the literature on human aggression, we view our analysis as a general purpose tool that can be useful for the broader research community (see link below for software download). Thus, we discuss the general features of our network citation approach below, along with the more specific implications for research on human aggression.

## 4.1 Implications for citation network analysis

Citation network analysis offers a considerably richer, data-driven characterization of the literature in comparison to focusing on citation count. Focusing on citation count alone emphasizes papers from the most popular community, but overlooks the community structure that may be present in the field. Indeed, we found that of the most cited papers, 70% belong to the largest community (media and video games), whereas many other communities including the second largest community (stress, traits, and aggression) were not represented at all by this approach. As such, citation ranking provides a lopsided perspective of the field as a whole. Alternatively, citation network analysis provides considerably more information that is useful for characterizing a field. Communities, influential articles per community, and bridging articles between communities—altogether provide a richly structured, topological map of a field.

Citation analysis may also help address the underrepresentation of women in contributions to science. Several studies across diverse research domains have reported evidence of gender differences in representation [8]. For instance, women appear to be underrepresented in influential publications and authorship positions [68, 69]. In certain fields, it has been reported that they receive about 10% fewer citations than men [70, 71]. In aggression research, focusing on citation count (or on the composite ranking) clearly shows a gender disparity in who gets cited in comparison to our citation analysis approach (Fig 5). Our citation analysis might therefore be useful for further identifying and crediting influential contributions by women scholars that may have been overlooked.

Citation network analysis may be useful for scientific pedagogy. Students and instructors may use the structure and indications of influential articles to curate topics and reading materials (e.g., for a syllabus) and ultimately guide scholarship development. With this goal in mind, we constrained our detailed output to the top five papers from the top 15 communities, which may provide a first approximation at establishing a balance of breadth and depth of a given field. Moreover, bridging papers may be useful for guiding scholarship development for integrating knowledge between fields. To be clear, we are not suggesting that citation network analysis should serve as a replacement from guidance by an expert in the field. Rather, it may be more useful as a supplement. Still, not all students or early career researchers in psychology have the privilege of an involved mentor, and so citation network analysis may help reduce barriers to entry concerning scholarship in the field. Relatedly, our network citation analysis may also be useful for the selection of topics and papers in conferences. It may be used to identify communities of interest, increase the diversity of topics, and help identify scholars based on influential articles including women scholars who may be underrepresented in certain areas. Further, our analysis characterized the timeline and growth of certain fields, which may also be useful for developing program content.

## 4.2 Guidelines for future usage

Citation network analysis provides a unique, data-driven method to characterize the topology of a research field. Here, we outline certain points of consideration for future usage. The first question concerns how to construct the citation matrix. This is perhaps the most critical step

since it holds the most influence over the literature map. Generating a citation matrix is non-trivial; most literature search engines do not provide this information directly. Here, we provide open source code for extracting a citation matrix using an API with Semantic Scholar. A citation matrix can also be developed using a variety of different approaches. We used a seed based approach combined with a reference trail (i.e. using reference lists to identify papers to include), which had advantages for our purposes in approaching the literature on human aggression (as outlined in the introduction) and provided a useful opportunity for validation (i.e. by examining whether we identified research communities similar to those characterized as topics in the review article). More generally, seed papers can be selected via quantitative criteria such as centrality, citations, journal impact factor, or by using qualitative criteria (e.g., expertise of the research team, perceived journal quality, reputation of a lead author) or a combination of both criteria [29]. Each approach has strengths and limitations that will ultimately impact the citation matrix [72–77].

Of course, it may be difficult to select a seed paper when one is new to a field. In such instances, it may be useful to consult other researchers who publish frequently on the topic if only to provide guidance on seed selection. However, there are certain guidelines that may be followed with, or without, substantial field expertise. Specifically, users may consider selecting a seed article using the same rationale that we outlined above: a highly-cited, peer-reviewed, comprehensive review article as are commonly published in certain outlets (e.g. the Annual Review series). To be sure, exactly which features matter will depend on the goals of the researcher in conducting their literature review. Relatedly, interpretation of the results should also be understood with respect to the strengths and limitations of the selected seed.

In lieu of a seed article, a citation matrix can be generated from keywords to identify articles and using citations to link the articles. A citation matrix defined from keywords may reveal communities that actually have no link to one another at all (whereas in seed-based approach every article is eventually linked to every other article in the network at least through the seed). Going one step further, keyword similarities can also be used as edges to link articles to one another instead of using citations [78–80] or even using the full-text to identify topics and communities (e.g., Liu et al., 2014 [81]). However, analysis using keywords or particular terms introduces its own set of challenges. For example, keywords can be expressed in different ways by the authors (e.g. synonyms), and so identifying relevant structures amongst keywords may require field expertise. Overall, we view these methods as complementary approaches in the broader family of literature network analysis. It may be of interest to future work to utilize a variety of different approaches to more formally compare and contrast them against each other and see how it changes the network model. Ultimately, however, the method one chooses depends on the goals of the researcher. Technically, our graph theoretic analysis can be flexibly applied to literature matrices that use a seed-based, keyword, or full-text approach. And our seed-based approach is particularly useful when expertise is minimal, so long as a seed can be identified perhaps by using the aforementioned strategies. Of note, it is also useful from a historical perspective for tracing the contributions of a particular line of work or author. Finally, there is no intrinsic limit to the seed-based approach; researchers can, if it is of interest to the goals of the researchers, generate citation matrices from different seeds and attempt to compare and contrast the resulting networks.

### 4.3 A Literature topology of research on human aggression

By applying our citation network analysis to research on human aggression, we provide the first literature map of this field. Our analysis identified multiple communities of aggression (Fig 2), their interconnectedness to one another (Fig 3), the growth of these communities (Fig

4), and a set of influential and bridging articles for each community (Table 3). While we analytically validated our method using mean NMI, it is also notable that our network citation method identified similar communities as delineated by the influential review paper by Anderson and Bushman [30]. For example, anger and aggression, alcohol and aggression, personality traits and aggression, and media and aggression were all described in distinct sections of the seed paper and they were also identified in our citation network analysis. Notably, while the seed paper reviews articles prior to its publication, our method only gathered papers that were published putatively after the seed paper (2002–2019). It is evident that the seed mostly focuses on work prior to 2002, whereas our approach takes into account articles published in the last two decades. Hence, our approach is not limited to confirming the review findings but also illustrates the development of these communities while identifying more recent communities in the field (e.g., testosterone and aggression; oxytocin), along with their level of interconnectedness. The topics of the communities described in our network varied widely ranging from aggression in media and video games (the largest identified community) to aggression in horses (the smallest identified community). We briefly discuss the content of these communities in the network. We limit ourselves to briefly discussing the top five communities since our aim is primarily to highlight the utility of the approach rather than to conduct a comprehensive review of the literature on human aggression.

In our largest community, the focus is on the effect that media violence has in increasing the likelihood of aggressive behavior, especially in youth [40]. We also find instances of cross-cultural evidence of how playing prosocial games increases prosocial behaviors [45], as well as explanations of the mechanisms through which exposure to violent video games causes increased levels of aggression [50]. The second-largest community is characterized by the theme of stress, personality traits, and aggression, and presents examples of how aggressive personality traits can predict support for political violence [51], as well as models of personality that might be employed to understand human aggression (e.g., [56]), and it reviews the role of stress on promoting prosocial decisions [61]. The third community focuses on rumination and displaced aggression; this community includes papers regarding how ruminating on a provocation may increase the likelihood of displaced aggression if a trigger is present [43], as well as papers on the trait of displaced aggression and its ability to predict indirect indicators of real-world displaced aggression [57]. The fourth community of our top five is centered on testosterone. Here, instances of how, in men, situational changes in testosterone modulate aggressive behaviors are presented [43], along with explanations of the possible effect of testosterone on aggression; this effect was interpreted with a reduction of the activity in the medial orbitofrontal cortex [53]. Finally, the fifth community focuses on social aggression and includes empirical research investigating not only physical but also nonphysical forms of aggression. For example, Heilbron, and Prinstein [59] reported that conceptualizing the effect of socially aggressive behavior at the individual, dyad, and group level might provide significant insights into factors that possibly play a role in maintaining and promoting social aggression. This community also includes neuroimaging studies where youth with aggressive conduct disorder show an atypical pattern of neural response when viewing other people's pain i.e., [44].

Our approach is not without some limitations. One limitation is that scholars may disagree on the assignment of specific papers to certain communities. As an example, the 10th most influential paper in the network, "Understanding impulsive aggression: Angry rumination and reduced self-control capacity are mechanisms underlying the provocation-aggression relationship [67]", was placed by the Louvain algorithm into the community entitled "media and videogames", however, if the community categorization was inferred solely by the paper title, this paper would have belonged to the community "rumination and displaced aggression"

instead. Such discrepancies may occur because we took a data-driven, unsupervised clustering approach to assign papers to communities. These assignments should not be over-interpreted as presenting the ground truth. Though we validated the network clustering results, which showed reasonably high consistency, adding more papers or starting from another seed could produce a different solution. Given these concerns, we urge readers not to view citation network analysis as a ground-truth representation of a field. Rather, it is a tool that provides a gist level representation of a field and guidance on how to approach it for scholarship training.

There were also some surprising omissions from our approach. For instance, there are relatively large research agendas dedicated to the developmental antecedents and consequences of aggression that did not appear to organize into a community in our analysis. Here, our analysis is limited in that it only utilized information regarding citations between papers. We did not include information about the scientific content (e.g., words used in the articles) from those papers in our clustering analysis. Future work using algorithms designed to handle text from included articles (e.g., using topic modeling) would assign papers to communities in a content-rich manner rather than our more content-free method that relies on citation links alone. Care must also be taken to limit over-interpretation of community size (as in Table 2). Certain communities we identified might have links to relatively large research communities of which aggression is an important aspect but is not the only focus (e.g., on PTSD). And certain communities may be larger than captured by our analysis. For example, alcohol and aggression, which is ranked eleventh with regard to the number of papers assigned to that community, likely involves a much larger community. Similarly, the size of a community should not be taken as evidence of the robustness of the research findings in the literature. For example, testosterone and aggression is the fourth-largest community, yet the relationship of testosterone to human aggression is complicated and weak [82]. Finally, our matrix included only 2 generations, and so the results are limited in addressing the prospective contributions of younger articles. It would be interesting to re-run the analysis in 5 years to see the evolution of the network and the emergence of new communities.

## 4.4 Conclusion and future directions

In conclusion, our study should be viewed as providing a first-pass, gist-level topological representation of the human aggression research using a data-driven, citation analysis approach. The analysis identifies many (but not all) communities in the field of aggression and captures many (but not all) influential articles in those communities. This literature topology may serve many purposes. It may provide a useful supplement for formal education and professional mentorship for learning about aggression while noting, of course, that it is not a replacement for either. Approaching the literature with citation network analysis enables researchers to quickly identify research communities and their influential papers within and across the human aggression field, which may be a critical tool for making future scientific advancements given the acceleration of research output. Finally, our network analysis may take one step to promote gender equity in aggression research by conditioning influential contributions by research communities. There are also other computational developments underway that may integrate well with our citation network analysis approach. It would be interesting, for example, to study the evolution of the network and how connections form within and across communities. It may also be possible, using machine learning approaches (e.g., [83]), to predict which papers will become relevant within the network and how different areas coalesce or diverge [84]. Doing so may help support continuous improvement in education by identifying trends and evolutionary trajectories [85], as well as assist in monitoring faculty development [86].

A possible limitation of this analysis could be that we only took into consideration the papers that cite the seed articles (first generation) and the papers that cite the first generation articles. Therefore, our network is composed of two levels of distance from the seed node. This structure of the network is likely to have produced an overlap in several metrics that we have chosen to calculate our composite scores and thus the communities. For instance, for several nodes (especially for the nodes with a larger number of edges) the value of betweenness centrality was very similar to the value of closeness. Potentially, this metric overlap could have brought to an overlap of the communities themselves, making it very difficult to discern them. However, it is crucial to remember that our validation analysis showed a good consistency among repetitions regarding community structure and the number of communities.

## Supporting information

**S1 File. Supplementary material.** This file contains the following sections. Section A: list of papers in the composite ranking, ranked from 11th to 20th position; Section B: list of the top 5 papers for the communities from the 6th to the 15th; Section C: top 10 papers based on citation ranking; Section D: principal component analysis (PCA) on network metrics; Section E: network metrics. The file contains the following figures and tables. Table 5: explanation of the variance of the principal component analysis on the network metrics. Fig 6: scree plot and PCA plot. Fig 7: representation of the top 10 papers for each network metric.
(PDF)

## Acknowledgments

We thank N. R. Rypkema for helping with data gathering and validation, Caitlyn Christiano, Rasvitha Nandru, and Zhaolin Li for their support in the gender analyses and manuscript preparation, and members of the Affective and Brain Sciences Lab at Northeastern University for feedback on earlier versions of this manuscript.

## Author Contributions

**Conceptualization:** Alessia Iancarelli, Thomas F. Denson, Ajay B. Satpute.

**Data curation:** Alessia Iancarelli.

**Formal analysis:** Alessia Iancarelli.

**Investigation:** Alessia Iancarelli, Thomas F. Denson, Ajay B. Satpute.

**Methodology:** Alessia Iancarelli, Chun-An Chou, Ajay B. Satpute.

**Project administration:** Alessia Iancarelli, Ajay B. Satpute.

**Resources:** Alessia Iancarelli, Thomas F. Denson, Chun-An Chou, Ajay B. Satpute.

**Software:** Alessia Iancarelli.

**Supervision:** Ajay B. Satpute.

**Validation:** Alessia Iancarelli, Chun-An Chou.

**Visualization:** Alessia Iancarelli.

**Writing – original draft:** Alessia Iancarelli.

**Writing – review & editing:** Thomas F. Denson, Chun-An Chou, Ajay B. Satpute.

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
