## [Decision Letter · Decision Letter 0]

8 Oct 2021

PONE-D-21-20644

A Network Approach to the Human Aggression Literature

PLOS ONE

Dear Dr. iancarelli,

Thank you for submitting your manuscript to PLOS ONE. After careful consideration, we feel that it has merit but does not fully meet PLOS ONE’s publication criteria as it currently stands. Therefore, we invite you to submit a revised version of the manuscript that addresses the points raised during the review process.

We look forward to receiving your revised manuscript.

Kind regards,

Hocine Cherifi

Academic Editor

PLOS ONE

Additional Editor Comments (if provided):

Reviewers' comments:

Reviewer's Responses to Questions

**Comments to the Author**

1. Is the manuscript technically sound, and do the data support the conclusions?

Reviewer #1: Yes

Reviewer #2: Yes

2. Has the statistical analysis been performed appropriately and rigorously? 

Reviewer #1: Yes

Reviewer #2: Yes

3. Have the authors made all data underlying the findings in their manuscript fully available?

Reviewer #1: Yes

Reviewer #2: Yes

4. Is the manuscript presented in an intelligible fashion and written in standard English?

Reviewer #1: Yes

Reviewer #2: Yes

5. Review Comments to the Author

Reviewer #1: General comments

This manuscript applies a community detection algorithm to analyze and identify influential papers per community of a citation network in the human aggression literature. It also analyzes possible gender biases of influential papers if communities of subfields are not considered.

Overall, I found the paper sound and original in the sense of applying well-established methods of network analysis to a particular citation network. However, before recommending for publication, I would like the authors to do a minor revision following the comments below.

Detailed comments

Introduction

• The narrative around the motivation for this study sometimes sounds like the method developed in the paper (and the problem of increasing publication volume) is specific to the aggression literature. It would sound better to pose the question in these lines: "How are scientists able to develop their expertise in a particular field, given the volume of research being published? Here, we propose a strategy – applied to the human aggression literature – to help researchers to acquire a breadth of understanding of the field of interest."

• Scientometrics (and citation networks in science) is yet another field that has increasing publication volume. The paper lacks engagement with the literature. How does your methodology fit in the literature of citation networks? Are you applying methods seen elsewhere? Are you adapting the methodology? How does it differ from others in the literature? I would like to see engagement with the literature other than "previous works have used citation network analysis to explore diverse scientific fields..."

• Paragraph 4. What biases and advantages? One example is the "positive bias for contemporary studies". However, I think a deeper reflection on possible biases and advantages might be helpful to other researchers that might want to use this method in their particular fields.

• Paragraph 4. How was the seed paper selected? Is it a subject choice based on one's knowledge of the field? The goal of the strategy proposed here is to help new researchers in the field navigating the literature. How can this new researcher choose the seed paper? How the results would change if another influential paper were selected. It would be good if the authors could test the robustness of the method subject to the choice of the seed paper.

Methodology

Section 3.1

• This is more of a reflection: Why just the first and second generation papers in the network? Couldn't it be the case that one of the poorly cited papers that were filtered out was just not cited by these two generations? That is, couldn't such a paper be opening an entirely new avenue, in a way that papers citing it do not cite the others in these two generations neither the source?

Section 3.2

• The third and fourth paragraphs of section 3.2 are confusing. Please standardize the notation and acronyms, and change the text to improve clarity.

Results

Section 4.1

• Have you tried other community detection methods? It would be nice to test how robust the results are subject to that as well.

• Figure 2. Would you please inform which algorithm was used for the layout of the network (force-directed?)? Please clarify if the size of the nodes is only to highlight the most influential ones. That is, are all other nodes of the same size?

• The first paragraph of page 9 is confusing and has repeated sentences. Please edit and clarify.

Section 4.2

• Table 3 seems to have no reference in the text where it appears. You only talk about this table in Section 5.

Section 4.3

• Fig 3 seems unnecessary, thanks to the total overlap between the bridging papers and the most influential by community, such that we can't extract any meaningful information from it. The paper doesn't explore this figure either, and Fig 2 already does the job.

• Do the results around more connected/isolated communities tell more about the field in general? I might have missed it, but I think there is no discussion about this even later on in Section 5.

Section 4.3.1

• I am a little skeptical with Figure 4. Please provide confidence interval and p-value.

• This entire section is somewhat confusing. Please rewrite to make your point (at the very end of the section) clearer.

Reviewer #2: The work is interesting and of importance to the relevant community. It is well written and the technical aspect is good (apart of 1-2 issues that will be shortly discussed).

The main claim of the authors is that network analysis is important since otherwise people rely solely on citation index. The evaluation of the paper also takes that into account and the results are taken against a citation ranking approach. However, the authors ignore the fact that people in a specific community also look for keywords that adhere to the topic, and hence will find relevant per-community (topic) papers when looking by keywords. This is an important distinction, which does not decrease from the importance of the current work. In addition, the authors have a directed graph which is temporal in nature, but seem to ignore it. I suggest that the work would also be suggested as a method for understanding the dynamics of the aggression research area. Which community stems from which and through which paper, and what's its growth projection. Please see detail remarks ahead.

In the chosen network analysis metrics there was little consideration to the directed nature of the networks. This is especially important in the case of the bridging nodes. The authors use an algorithm for finding bridging nodes that considers an undirected network. Following this disregard for the directed nature of the network, the conclusions from the results assume a bi-directional network. The authors say that "the top five papers in the top five communities are not only the most influential within their communities but they are also connecting papers between communities in the human aggression network". This conclusion is valid Only for an undirected network. For a directed network, I would imagine that these papers, building on "older" communities, maybe found a new angle that became the basis for the newer community.

- Please see to discuss the effect of the network being directed on the bridging algorithm and reconsider your conclusions in this regard.

Another thing that I have noticed is that about third of the prominent papers depicted in Figure 7 in the supporting information came from the largest community (apart from the case of figure 7d, the bottleneck papers). Is this community the "Media and Video games"? It is the largest in the network, and the denser. It is easy to conclude that high degree nodes are then highly cited within the community and also from other communities, hence have high degree and also high closeness (and of course high EPC and high MNC).

It should then be noted that the choice of having only two levels of distance from the seed node caused an overlap in the majority of the chosen metrics.

- Please refer in your text to the points of overlap in the metrics due to the structure of the network and the resulted overlap in some of the results, especially for prominent papers.

The directed nature of the network implies also temporal ordering. This means that younger papers have less chance of being cited than older ones. Is it possible to control for this in some way? I am not from the Regression world but would it be safe to say that the size of some of the communities correlated with their age?

- If so, an interesting thing to find is "growing communities" vs. "stable communities". One might think of it maybe as "trendy communities" = find for each community the fraction of the number of new citations to their size, and see which is the fastest growing and whether there are communities that are starting to become less appealing?

Lastly, the entire methodology relies on your understanding of the research area, by choosing a seed paper. This could be the reason to some of the limitations you discuss. I suggest to discuss alternatives to the seed approach in the Methodology Section and their implications.

Minor remarks:

1. Methodology chapter, third sentence: refers  refer

2. Page 9, first paragraph - some repetition there, it needs editing.

3. Section 4.2, "As an example... : - I really don't understand what is written there. You named communities by the prominent papers in it. What is the meaning of this "community categorization inferred solely by the paper title"?

6. PLOS authors have the option to publish the peer review history of their article (what does this mean?). If published, this will include your full peer review and any attached files.

Reviewer #1: **Yes: **Demival Vasques Filho

Reviewer #2: **Yes: **Osnat Mokryn

---

## [Author Response · Author response to Decision Letter 0]

8 Feb 2022

The responses to Editor and Reviewers have been uploaded as an external document "Response_to_Reviewers.pdf"

---

## [Decision Letter · Decision Letter 1]

23 Mar 2022

Using Citation Network Analysis to Enhance Scholarship in Psychological Science: A Case Study of the Human Aggression Literature

PONE-D-21-20644R1

Dear Dr. iancarelli,

We’re pleased to inform you that your manuscript has been judged scientifically suitable for publication and will be formally accepted for publication once it meets all outstanding technical requirements.

Kind regards,

Hocine Cherifi

Academic Editor

PLOS ONE

Additional Editor Comments (optional):

Reviewers' comments:

Reviewer's Responses to Questions

**Comments to the Author**

1. If the authors have adequately addressed your comments raised in a previous round of review and you feel that this manuscript is now acceptable for publication, you may indicate that here to bypass the “Comments to the Author” section, enter your conflict of interest statement in the “Confidential to Editor” section, and submit your "Accept" recommendation.

Reviewer #2: All comments have been addressed

2. Is the manuscript technically sound, and do the data support the conclusions?

Reviewer #2: Yes

3. Has the statistical analysis been performed appropriately and rigorously? 

Reviewer #2: Yes

4. Have the authors made all data underlying the findings in their manuscript fully available?

Reviewer #2: Yes

5. Is the manuscript presented in an intelligible fashion and written in standard English?

Reviewer #2: Yes

6. Review Comments to the Author

Reviewer #2: Hi

All my review comments from the first round were answered thoroughly.

I requested a Minor Revision as I find it challenging to understand Figure 4.

The presentation is confusing, and it is unclear why there are no papers in 2019? Maybe create parallel bars or find another way to represent the longitudinal information. Please explain the figure better.

Also, I am curious - can you qualitatively explain what's happened around 2008? A massive boost in publications / new communities around the field (but for the stress community, which is also interesting)?

I do not require that you check it. I was just curious.

7. PLOS authors have the option to publish the peer review history of their article (what does this mean?). If published, this will include your full peer review and any attached files.

Reviewer #2: **Yes: **Osnat Mokryn

---

## [Editor Report · Acceptance letter]

28 Mar 2022

PONE-D-21-20644R1 

Using Citation Network Analysis to Enhance Scholarship in Psychological Science: A Case Study of the Human Aggression Literature 

Dear Dr. Iancarelli:

I'm pleased to inform you that your manuscript has been deemed suitable for publication in PLOS ONE. Congratulations! Your manuscript is now with our production department. 

Kind regards, 

on behalf of

Professor Hocine Cherifi 

Academic Editor

PLOS ONE